# A colloidal viewpoint on the sausage catastrophe and the finite sphere packing problem

Susana Marín-Aguilar [1] ✉, Fabrizio Camerin [1,2] ✉, Stijn van der Ham [3], Andréa Feasson[3], Hanumantha Rao Vutukuri [3] ✉ & Marjolein Dijkstra [1,2] ✉

It is commonly believed that the most efficient way to pack a finite number of equal-sized spheres is by arranging them tightly in a cluster. However, mathematicians have conjectured that a linear arrangement may actually result in the densest packing. Here, our combined experimental and simulation study provides a physical realization of the finite sphere packing problem by studying arrangements of colloids in a flaccid lipid vesicle. We map out a state diagram displaying linear, planar, and cluster conformations of spheres, as well as bistable states which alternate between cluster-plate and plate-linear conformations due to membrane fluctuations. Finally, by systematically analyzing truncated polyhedral packings, we identify clusters of $56 \leq N \leq 70$ number of spheres, excluding $N = 57$ and 63, that pack more efficiently than linear arrangements.

The best way of packing spheres has a long history, dating back to the works of Kepler, Gauss, and Newton, while the British sailor Raleigh was also intrigued by this problem as he searched for an efficient way to stack cannonballs on his ship[1]. Sphere packings also have applications in coding theory, wet computing, crystallography, and in understanding mechanical and geometrical properties of materials[2–7]. In 1611, Kepler conjectured that the densest packing of an *infinite* number of identical, non-overlapping spheres in three-dimensional (bulk) conditions is the "cannonball" stacking or the face-centered cubic (FCC) crystal, which fills space with an efficiency of ~74%. This hypothesis was proven mathematically only recently[8,9].

In reality, however, all packings are inherently *finite*, which means that their extension is limited in space. This raises the question of what is the most efficient way to pack equally-sized spheres in either a container with a predefined shape[10–12], or inside a flexible container like the smallest convex hull that encloses the spheres[13]. Surprisingly, the densest packing of a number of spheres $N$ within their convex hull is not always a compact cluster of spheres. On the contrary, in 1975 the mathematician Fejes Tóth conjectured[14] that for dimensions $d \geq 5$ the densest packing is the one where the centers of the spheres are aligned along a straight line, resulting in a so-called sausage configuration. This conjecture, supported by other studies [15,16], was initially proven true only for $d \geq 13387$[17] and subsequently for $d \geq 42$[18]. No proof of its general validity has thus been reported for lower dimensions.

For $d = 4$, a sudden transition in the packing density occurs from a linear to a cluster arrangement, where the coordinates of the particles extend in all three dimensions, and is typically referred to as the "sausage catastrophe"[19,20]. The upper bound of $N = 375769$ that was initially assigned to this transition[21,22] has been recently reduced to $N = 338196$[23]. For $d = 3$, different studies reported that the sausage conformation minimizes the volume of the convex hull for $N \leq 55$, and for $N = 57, 58, 63, 64$[19,24,25], while above this limit the densest configuration becomes a three-dimensional cluster, thereby avoiding the plate conformation where the centers of the spheres are positioned on a plane[15,26]. However, the precise structure of these clusters, which are denser than the sausage, remains largely unknown. Furthermore, there may be other unidentified clusters that present an even denser packing than the linear arrangement.

[1]Soft Condensed Matter & Biophysics, Debye Institute for Nanomaterials Science, Utrecht University, Princetonplein 1, Utrecht 3584 CC Utrecht, The Netherlands. [2]International Institute for Sustainability with Knotted Chiral Meta Matter (WPI-SKCM²), Hiroshima University, 1-3-1 Kagamiyama, Higashi-Hiroshima 739-8526 Hiroshima, Japan. [3]Active Soft Matter and Bio-inspired Materials Lab, Faculty of Science and Technology, MESA+ Institute, University of Twente, 7500 AE Enschede, The Netherlands. ✉e-mail: s.marinaguilar@uu.nl; f.camerin@uu.nl; h.r.vutukuri@utwente.nl; m.dijkstra@uu.nl

Despite its fundamental importance, the finite sphere packing problem has primarily been studied from a mathematical perspective, and making an experimental realization, even for a limited number of spheres, is still a significant challenge. On the other hand, colloidal hard spheres serve as an ideal model system for investigating particle packings[27–29]. By leveraging their excluded-volume interactions and using an appropriate flexible container, we can explore the various conformations that the spheres can adopt. In our study, we employ giant unilamellar vesicles (GUVs), which are effectively large elastic containers in which colloidal particles can be enclosed[13,30–33]. The dynamics of GUVs can be studied using confocal microscopy, enabling direct observation of their shape fluctuations[32,34,35]. Furthermore, GUVs possess the ability to alter their shape in response to external stimuli such as changes in osmotic pressure[36–38] and forces exerted by passive[39] and active particles[32,40,41]. However, experimental realizations of sausage- and plate-like arrangements remain elusive.

Here, we examine how membrane fluctuations, induced by Brownian colloidal spheres inside the vesicle, affect the different vesicle and colloid conformations. Combining experiments on silica or polystyrene particles enclosed in GUVs with computer simulations, we demonstrate that colloidal spheres can form stable linear, plate, and cluster conformations under certain physical conditions. We summarize our results for $N \le 9$ in a state diagram, which displays the aforementioned conformations of spheres as a function of a single-order parameter, allowing us to extract information about their packing. Additionally, we discover bistable states in which the system alternates between cluster-plate and plate-linear conformations. We then determine whether the sausage catastrophe can be observed in finite systems where the spheres are not close-packed, and that accounts for the entropy of the positional degrees of freedom of the colloidal particles. Finally, we identify the conditions required to form finite clusters with high packing efficiency for a large number of spheres and study them systematically. In this way, we uncover clusters composed of $N = 58$ and 64 spheres that exhibit better packing than the linear conformation. As a result, we provide evidence for the existence of particle arrangements with higher packing efficiency compared to those previously examined[19,25], thereby lending direct support to Fejes Tóth original conjecture.

## Results and discussion

We begin our study by exploring the conformations of $N$ colloids confined in a fluid vesicle both in experiments and simulations. In experiments, we use a modified droplet transfer method[34] to encapsulate colloidal particles of size 2.12 $\mu$m in GUVs, drawing inspiration from a previous work by Vutukuri, et al.[32]. Next, the vesicles are exposed to hypertonic shock, where the solute concentration outside the vesicle is higher than inside, in order to control their morphology (see Methods). The vesicle morphologies and particle dynamics are followed by a fast confocal scanning microscope. We note that, while ref. 32 dealt with self-propelled particles locally deforming the lipid membrane, here we entirely focus on passive particles. Furthermore, the vesicles employed in this study significantly differ from oil emulsion droplets[10], whose surface tension is several order of magnitude higher than that of GUVs.

In the molecular dynamics simulations, various vesicle shapes are investigated using a meshless membrane model[42,43]. In this model, lipids are represented in a coarse-grained fashion using spheres of diameter $\sigma$, which is also used as the unit of length in our simulations. The model incorporates orientation-dependent interactions to account for the properties and interactions of the constituent lipids in real GUVs. The membrane is designed to have an approximately null surface tension, similar to the experiments. We use explicit solvent to control the shape of the vesicle but, differently from the original model, we only add it to the outer region thus exerting an external pressure on the membrane. In each vesicle we insert a number of

colloids $N \in [3, 9]$ with a diameter of $\sigma_c = 12\sigma$. The colloids interact via a repulsive Weeks-Chandler-Andersen (WCA) potential. The use of a WCA potential instead of a hard-sphere potential enables the system to be treated with molecular dynamics simulations, which can be fully implemented into efficient simulation packages like LAMMPS[44]. Additional details on the simulations and the interaction potentials are provided in the Methods section.

In both experiments and simulations, the size (and therefore the surface area) of the vesicle is adjusted depending on $N$. We observe that state points obtained from simulations and experiments encompass a wide range of packing fractions $\eta = NV_0/V_v \approx 0.12 - 0.28$ for the colloids in the vesicle (see Fig. S1), with $V_0$ representing the volume of a colloid and $V_v$ the volume of the vesicle. The latter is estimated by generating the surface mesh of the vesicle[45,46], which allows us to directly extract the value of the volume enclosed within the meshless membrane (see Supplementary Information). The range of investigated packing fractions should be related to $\eta \simeq 0.7$ of the tightest sausage configuration. To compare vesicles with different surface areas as obtained in simulations and experiments, we introduce the reduced volume $v$, which is the ratio between the volume of the vesicle $V_v$ and the volume of a sphere $V_s$ with the same surface area as the vesicle $A_v$:

$$\nu = \frac{V_v}{V_s} = 3\sqrt{4\pi}\frac{V_v}{A_v^{3/2}}. \tag{1}$$

The parameter $v$ takes values between $0 < v \le 1$, with $v = 1$ corresponding to a spherical vesicle[47]. From the definition of the reduced volume $v$, it can be deduced that vesicles with the same number of colloids $N$ and surface area $A_v$ will have a higher colloid packing fraction $\eta$ when the value $v$ is lower. To obtain $v$ in simulations, we determine the surface area of the vesicle, $A_v$, by constructing a surface mesh around it, similar to the approach used for $V_v$[45]. On the other hand, in experiments, these values are extracted from the $xyz$ confocal data using an ImageJ plugin (see Methods and Supplementary Information). Each colloid conformation is characterized by the anisotropic shape parameter $\kappa^2 = 3(a_x^2 + a_y^2 + a_z^2)/2(a_x + a_y + a_z)^2 - 1/2$[48], determined by calculating the eigenvalues $a_x, a_y, a_z$ of the diagonalized gyration tensor, which is constructed from the $x$, $y$, and $z$ coordinates of the colloidal particles. This proves particularly effective in distinguishing linear conformations, characterized by $\kappa^2 \gtrsim 0.5$, planar arrangements, indicated by $0.2 \lesssim \kappa^2 \lesssim 0.3$, and clusters with a more isotropic shape, represented by $\kappa^2 \approx 0$.

In Fig. 1(a), we summarize our results both from experiments and simulations in a state diagram as a function of $v$ and number of colloids $N \in [3, 9]$. We use different symbols to denote different colloid conformations as identified by $\kappa^2$, encoded in the color coding, and visual inspection. The orange dashed line represents the reduced volume corresponding to the optimal linear packing where all spheres are in contact, i.e. a spherocylinder of length $(N - 1)\sigma_c$ and radius $\sigma_c/2$ for which $v_{lin}(N) = (4/3 + 2(N-1))/(4/3N^{3/2})$. We note that by employing the reduced volume $v$, the state diagram becomes independent of vesicle size. For each $N$, we identify different regions denoted by different shades of blue where the linear, planar, and cluster configurations are prevalent. Figure 2 displays these arrangements, featuring composite bright-field and confocal microscope images, as well as simulation snapshots. Movie S1 shows the linear arrangement for $N = 9$ both for experiments and simulations. Upon increasing the number of colloids $N$, we find that the linear conformation becomes stable in a wider range of $v$, whereas for $N > 9$ elongated vesicles exhibited excessive bending in both experiments and simulations. In all cases, we find cluster conformations for $v > 0.9$, plate conformations for intermediate $v$, and linear arrangements for the lowest $v$. The state diagram shows good agreement between experimental findings and simulation results.

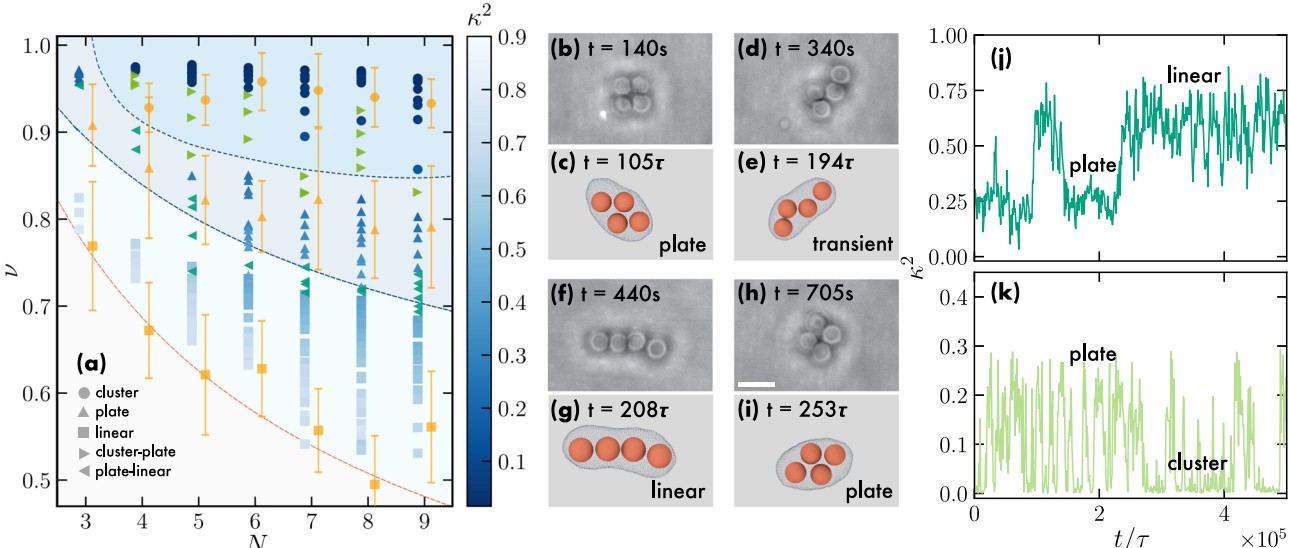

**Fig. 1 | Colloidal realization of the finite sphere packing problem. a** State diagram of colloidal hard spheres enclosed in GUVs as a function of the number of colloids $N$ and the reduced volume $v$. Different symbols denote different colloid arrangements, according to the legend. Blue-shaded symbols are numerical state points colored according to the anisotropy shape parameter $\kappa^2$, green symbols indicate numerical bistable state points, and orange symbols are experimental data. The error bars represent the standard deviation of the experimental data points combined with an image analysis error (see Supplementary Information). The orange line corresponds to $v_{lin}(N)$, while the blue lines are guides-to-the-eye for identifying the regions where linear, planar, and cluster configurations are predominant. **b–i** Sequence of time-lapse images obtained from 2D composite confocal and bright-field microscopy, and representative simulation snapshots revealing a bistable line-plate state point. The scale bar is $5\,\mu m$. **j, k** $\kappa^2$ as a function of simulation time $t/\tau$ for bistable plate-linear and plate-cluster states, respectively, where $\tau = \sqrt{m\sigma^2/\epsilon}$ represents the unit of time in simulations (see Methods).

Additionally, we find bistable regions due to the combined membrane, shape, and solvent fluctuations driven by the colloids inside the vesicle. These regions are identified in simulations by calculating the order parameter $\kappa^2$ as a function of time. Fig. 1(j,k) demonstrate bistability between linear-plate and plate-cluster conformations, respectively, for $N = 4$, where we can clearly see that the order parameter $\kappa^2$ fluctuates between the values corresponding to the different conformations. We denote these bistable state points as left and right green triangles in the state diagram in Fig. 1(a). Fig. 1(b–i) show time-lapse snapshots of a bistable state point in both experiments and simulations, where we observe that a vesicle in an initial planar configuration transitions to a linear conformation (Movie S2). The transition is reversible as the colloids return to a planar conformation at longer times. To highlight the robustness of our methods, we convincingly demonstrate the transition of the particles from a linear arrangement to a clustered state both in simulations and experiments. This transition is achieved by precisely controlling the surface area-to-volume ratio of the vesicle through osmotic imbalances across the membrane, as depicted in Fig. S3 and Movie S3.

As mentioned earlier, the reduced volume $v$ also provides insight in the packing behavior of the colloids when the surface area of the vesicles remains constant. Although it is challenging to meet this requirement in experiments at fixed $N$, in simulations the surface area values for the vesicles remain nearly constant for a given $N$ (see Table S1). As a result, our in silico realization of the finite sphere packing problem unequivocally confirms that packing efficiency is maximized in the linear conformation when compared to the plate or cluster arrangements, as evidenced by the consistently lower values of $v$ observed across all values of $N$. To emphasize this aspect, we present a state diagram in Fig. S1 that explicitly reports the packing fraction for the state points analyzed in the simulations. We also note that the packing efficiency significantly increases as the number of colloids enclosed within the flexible container increases.

In this way, we have determined the physical conditions that allow the observation of linear, planar, and cluster conformations of $N \in [3, 9]$ hard-sphere colloids in a flexible vesicle. However, the number of colloids is rather limited, not only due to the significant bending of the vesicles but also because of computational constraints. This number is notably lower than the number of colloids predicted to result in the sausage catastrophe, which is anticipated to occur at $N = 56$ for $d = 3$. We, therefore, investigate by means of simulations the possibility of observing the sausage catastrophe in a flexible vesicle and identifying cluster conformations of spheres that pack better than the linear arrangement.

To this end, we place $N \in [10, 150]$ colloids in a spherical vesicle that surrounds the colloids as tight as possible without breaking the layer of beads composing the vesicle, and resulting in a packing fraction $\eta \approx 0.4$. We perform simulations of the colloids in the vesicle, and collect the different cluster conformations. For these simulations we use the same parameters as those employed in the first part of this study, which are further described in the Methods section. The sizes of the vesicles used in the simulations are provided in Table S2. In addition, we also consider dense clusters taken from a database[49]. These clusters were obtained by minimizing the energy of systems composed of Lennard-Jones (LJ) particles. In this way, we have at our disposal a large variety of clusters of which we can assess their characteristics and investigate their packing. After implementing an optimization protocol to approach the hard-sphere limit, we determine the colloid packing fraction for both data sets as $\eta_{ch} = N V_0/V_{ch}$, where $V_{ch}$ is the volume of the convex hull that encloses the colloids. This numerical protocol complements the analysis just presented on the flexible vesicle. In fact, besides being able to explore states with a larger number of colloids, by using $\eta_{ch}$ we effectively study the packing fraction of the tightest possible container, and thus compare it to the ideal linear packing fraction $\eta_{lin}$ obtained from the volume of a spherocylinder with $N$ particles (see Supplementary Information). In contrast, achieving states with a very high packing fraction experimentally would be significantly more challenging. This is because it is not feasible to directly manipulate the volume-to-surface area ratio to tightly enclose the colloidal particles within the vesicles. Additional information regarding the optimization protocol and the construction of the convex hull can be found in the Supplementary Information.

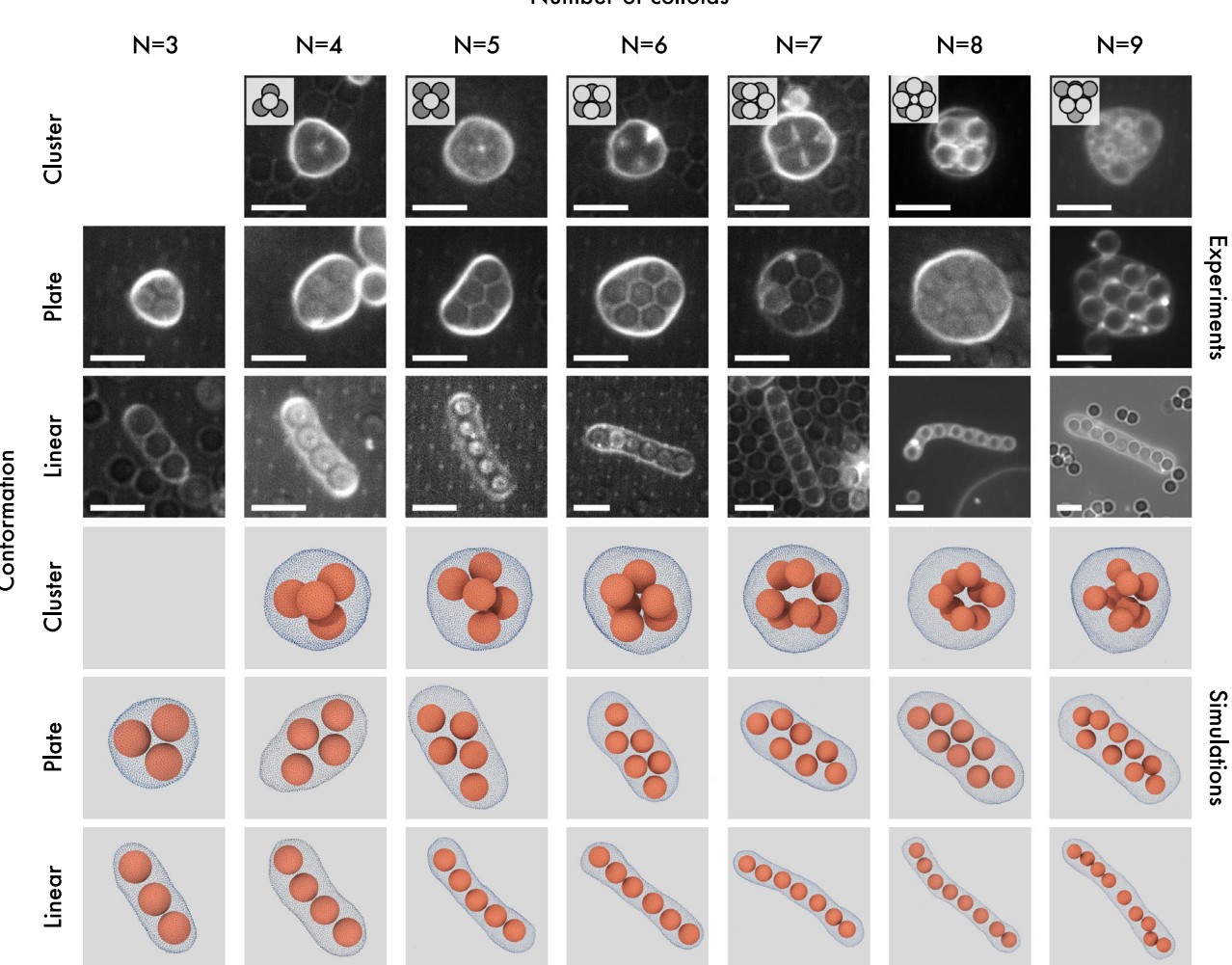

**Fig. 2 | Linear, planar and cluster conformations for colloids in a vesicle.** Experimental images obtained through a combination of bright-field and confocal microscopy, along with simulation snapshots, representing cluster, plate, and linear configurations. The insets provide schematic representations of the cluster arrangement observed in experiments. The scale bars in all experimental images are $5\,\mu m$.

We report the packing fraction $\eta_{ch}$ of the simulated clusters and those from the database in Fig. 3(a) as a function of $N$. We indicate with a vertical dashed line the number of particles $N$ at which a cluster, according to literature[19,20,25], is expected to exhibit a higher colloid packing fraction $\eta_{ch}$ than the linear conformation, which is referred to as the sausage catastrophe. For comparison, we also plot $\eta_{ch}$ of icosahedral clusters for varying $N$, as these structures are expected to pack locally very efficiently[12,50]. Surprisingly, we observe that all the clusters studied have a lower $\eta_{ch}$ than the linear conformation, with the exception of two icosahedra with $N > 100$, which have slightly higher packing fractions than the linear conformation. We thus find that only icosahedral clusters with particle numbers $N$ much larger than where we expect the sausage catastrophe to occur pack better than the linear conformation.

Additionally, we observe that the clusters obtained from simulations generally exhibit a similar colloid packing fraction $\eta_{ch}$ compared to the clusters from the database. However, it is worth noting that only a few clusters exhibit significantly higher packing fractions than the others, thus approaching the linear packing. As shown in Fig. 3(b), many of these clusters exhibit a high bond-orientational order parameter $q_6$[51] compared to the others. This indicates the presence of an underlying FCC crystalline order in their arrangement as $q_6$ serves as a measure of the six-fold symmetry within the cluster. As an example, let us consider the cluster with $N = 38$ taken from the database (see Figs. 3

and S8). This cluster has been previously identified as a minimum-energy configuration of the LJ potential and has been found to be stable for a wide range of LJ parameters[49,52]. This specific cluster has a truncated octahedral shape based on an FCC arrangement, which allows for the presence of regular two-dimensional patterns on its surfaces.

Based on these findings, we anticipate that clusters surpassing the linear packing will exhibit similar characteristics to the previously analyzed cluster. Specifically, we expect them to maintain an FCC structure and display regular patterns on their surfaces. Therefore, in the subsequent part of our study, we examine a number of ordered arrangements in which the spheres are in contact. Furthermore, we also present a similar analysis for other representative Barlow stacking arrangements of spheres[53–55], such as hexagonal close packing (HCP) (see the Supplementary Information). Our findings demonstrate that these arrangements generally provide less efficient packings compared to the FCC.

We start by analyzing the packing fraction $\eta_{ch}$ of two-dimensional planar structures, as reported in Fig. 4(a) together with the packing fraction of the linear arrangement, the sausage, for clusters with varying number of particles $N$. The triangular and hexagonal conformations exhibit packing fractions that approach that of the linear arrangement for small $N$. However, as $N$ increases, $\eta_{ch}$ decreases rapidly until it reaches a plateau at around $\eta_{ch} \approx 0.60$, which is lower

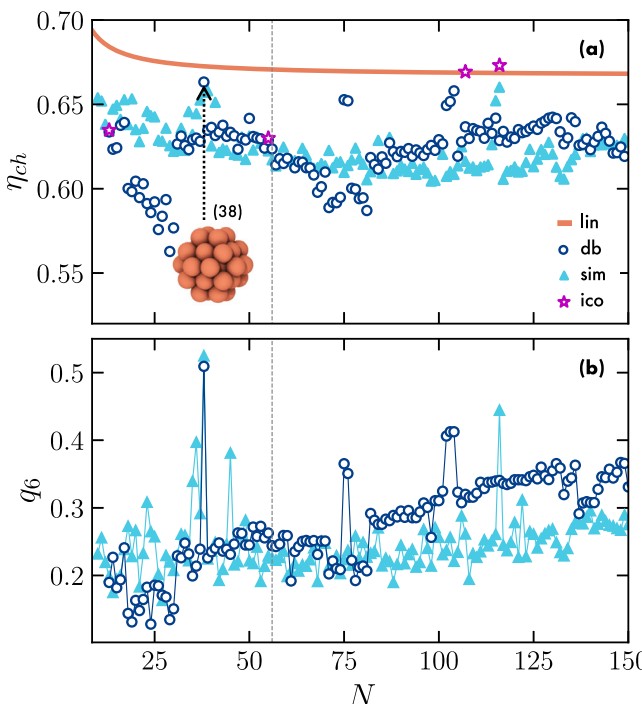

**Fig. 3 | Colloidal clusters enclosed in a vesicle. a** Packing fraction $\eta_{ch}$ of clusters composed of particles in the hard-sphere limit whose initial configurations were taken from simulations (*sim*, triangles) and from a database (*db*, circles), plotted as a function of the number of particles $N$. The packing fraction of the ideal linear conformation $\eta_{lin}(N)$ is depicted as an orange line (*lin*), along with the packing fraction of icosahedra (*ico*, stars). The inset snapshot displays the $N = 38$ cluster obtained from the database. **b** Bond-order parameter $q_6$ as a function of $N$ for the clusters as obtained from simulations and the database, as shown in (a).

than the packing fraction of the sausage conformation. As expected, a square arrangement of spheres display a significantly lower packing fraction than that of the triangular and hexagonal conformations, and thus also of the linear conformation. The fact that two-dimensional structures exhibit an $\eta_{ch}$ well below that of the linear arrangement is consistent with previous work which demonstrated that the optimal packing can only be achieved with either one- or three-dimensional conformations, i.e. in linear or cluster arrangements, respectively[15]. Nonetheless, based on the state diagram presented in Fig. 1(a), we expect that planar conformations would remain stable even for a large $N$.

Subsequently, we turn our attention to three-dimensional clusters. We start with some of the simplest polyhedra such as tetrahedra, octahedra and bipyramids. To explore a large variety of clusters, we consider polyhedra of different sizes constructed by slicing a close-packed FCC crystal, and perform subsequent cuts to the vertices of all shapes. We use the notation $X_n^k$, where $X = T, O, B$ for tetrahedra, octahedra or bipyramids, respectively, $n$ represents the number of particles removed from each vertex, and $k$ denotes the number of vertices from which particles are removed (see also Supplementary Information). We simply use $X_n$ for isotropic cuts.

In Fig. 4(b), we show the convex-hull packing fraction $\eta_{ch}$ and typical configurations for regular and for some sliced tetrahedra $T_1$, $T_4$, and $T_{10}$. For $N = 4$, $\eta_{ch}$ of the regular tetrahedron is very close, but slightly lower, to that of four spheres on a line. As the number of particles $N$ increases, the colloid packing fraction $\eta_{ch}$ initially decreases until it reaches a minimum value. Subsequently, it crosses the packing fraction of the line at $N = 84$, which is much higher than where we expect to find the sausage catastrophe. We observe that by removing particles from the vertices $\eta_{ch}$ increases, but the cluster with the lowest

number of particles that packs better than the sausage is the $T_4$ consisting of $N = 68$ colloids.

Similarly, Fig. 4(c) reports $\eta_{ch}$ for regular and regularly truncated octahedra $O_1$, $O_5$, and $O_{10}$ with corresponding configurations. In contrast to tetrahedra, octahedral clusters show a decrease in $\eta_{ch}$ as more particles are removed from the vertices, and the crossover to the packing fraction of a sausage occurs at larger $N$. In general, $\eta_{ch}$ for octahedra is lower than that of tetrahedra, particularly for $N < 50$. The truncated octahedra with the lowest number of particles that packs more efficient than the linear conformation is the $O_1$ with $N = 79$ spheres.

Subsequently, we analyze how the packing is affected by an asymmetric removal of spheres from regular tetrahedra $T$, octahedra $O$ and bipyramids $B$, i.e. by removing a different number of spheres from each vertex or layer of the polyhedra. Specifically, we focus on the region close to the value of $N$ where the sausage catastrophe was predicted to occur. We present the results in Fig. 4(d) as a function of $N$, with filled symbols denoting tetrahedra, octahedra and bypiramids whose packing fraction exceeds that of the linear arrangement (see also Figs. S9–S11 and interactive HTML files provided as Supplementary Data 1). While the mathematical requirements for tetrahedra and bipyramids to form clusters denser than the sausage have been established[25], such a prediction has not been made for octahedra. Remarkably, we observe that in this range of $N$ only polyhedral clusters that are sliced asymmetrically pack denser than the sausage. In particular, we note that two clusters, one with $N = 58$ and the other with $N = 64$, which were previously believed to have optimal packing in the linear conformation[25], actually exhibit better packing efficiency than the sausage conformation. Our results further indicate that regular or regularly cut polyhedra are generally not the arrangements that maximize the packing efficiency. Furthermore, we observe no specific correlations between different structural elements of the clusters that can directly influence packing, such as the number of faces, edges, or vertices (see Table S3). It appears that better packing results from nontrivial combinations of all these elements, with each contributing marginally to minimize the available volume.

In conclusion, our study sheds new light on the finite sphere packing problem and provides valuable insights on the most efficient methods for packing a finite number of spheres in a closed, flexible container. We demonstrate that a low-tension vesicle can serve as a model system for studying linear, planar, and cluster configurations formed by a small number of colloidal hard spheres. By constructing a general state diagram based on a single-order parameter describing the reduced volume of the vesicle, we can differentiate between stable and bistable states, and demonstrate how for such systems a linear arrangement of colloids always presents a better packing than other configurations. Our simulation predictions are consistent with the experimental observations. Beyond addressing the finite sphere packing problem, our findings have broader applications. For instance, the encapsulation of a limited number of spheres within a vesicle provides a strategy for pre-assembling building blocks that could be used to construct larger, more intricate structures[56,57], with potential applications such as the enhancement of plasmonic properties in metamaterials[58]. Moreover, our methodology can be adapted to other building blocks like dimers, trimers, or tetramers, drawing inspiration from existing techniques that use patchy interactions or emulsion methods to realize these clusters[10,59–61].

Subsequently, our simulations predict that simply packing spheres within a vesicle does not result in configurations with higher packing fractions than the linear arrangement, even when using a larger number of particles. However, upon closer examination of the structure and arrangement of the particles, we find that higher packing fractions can only be achieved with faceted, ordered clusters, such as with truncated tetrahedra, octahedra, or bipyramids in the region where the sausage catastrophe occurs. Our systematic investigation of

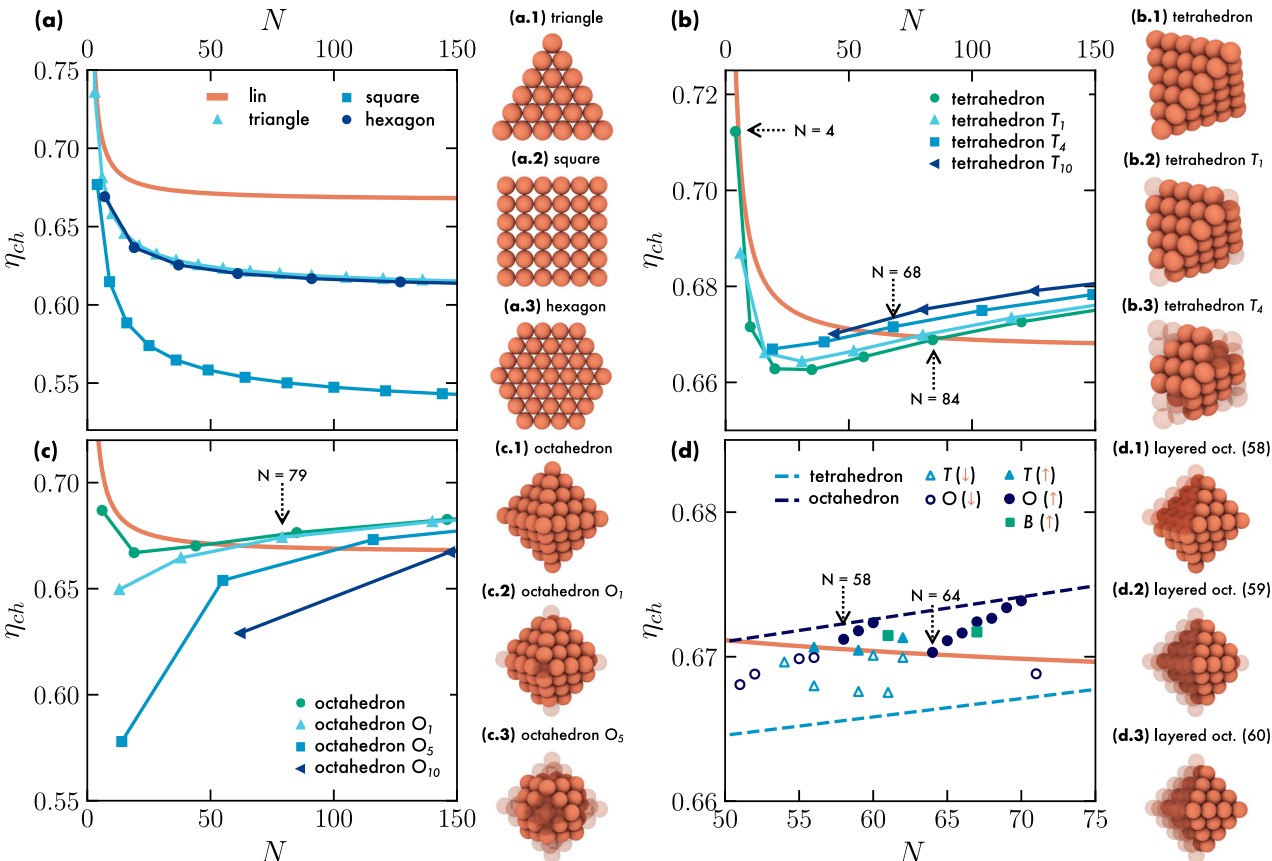

**Fig. 4 | Packing of ordered arrangements of colloids.** Packing fraction $\eta_{ch}$ of clusters consisting of spheres enclosed within their convex hull as a function of the number of particles $N$ for **a** planar arrangements, **b** regular and regularly truncated tetrahedra $T_n$, with $n$ the number of particles removed from each vertex, **c** regular and regularly truncated octahedra $O_n$, and **d** for a range of irregularly truncated tetrahedra $T$, octahedra $O$ and bipyramids $B$, as indicated in the legends in each panel. The orange line represents the packing fraction of the linear arrangement of spheres $\eta_{lin}(N)$. In **d**, open symbols represent irregularly truncated tetrahedra and octahedra with $\eta_{ch} < \eta_{lin}$ ($\downarrow$), while closed symbols indicate cases where $\eta_{ch} > \eta_{lin}$ ($\uparrow$), including bipyramids. Each panel is accompanied by example clusters, where the original non-truncated configuration is shown with transparent spheres. Dotted arrow lines are used to facilitate the identification of certain clusters that are mentioned in the text.

these clusters with various shapes has allowed us to directly prove, with a practical approach inspired by the physics of colloids, the existence of previously unidentified clusters that exhibit a packing efficiency that is superior to the sausage configuration, highlighting how finite sphere packing is still an open and intriguing problem. Nevertheless, it remains to be determined whether mathematical proofs can be developed for the packing of these clusters and for the entire Fejes Tóth conjecture. We believe that our work can serve as a catalyst for further research in this direction. From a computational perspective, we envision the development of cluster generation techniques, either through conventional or machine-learning methods[62]. Such approaches could expand the exploration of even more configurations and different Barlow stacking arrangements of spheres. Finally, while creating these clusters by enclosing spheres within a flexible vesicle may not be feasible, they could potentially be observed in systems characterized by strong attractive interactions, such as in gold nanoparticles and platinum clusters [63–66]. Experimental investigations into these complexes, as described in this work, have the potential to provide a definitive realization and demonstration pertaining to the fascinating problem of finite sphere packing.

## Methods
### Experimental section
**Materials.** All chemicals, unless otherwise specified, were used as received. 1,2-dioleoyl-sn-glycero-3-phosphocholine (DOPC) and

fluorescent 1,2-dioleoyl-sn-glycero-3-phosphoethanolamine-N-(lissamine rhodamine B sulfonyl) (ammonium salt) (Liss Rhod PE) in chloroform were obtained from Avanti Polar Lipids (Alabaster, AL). Chloroform (≥99.5%), paraffin oil, (heavy) mineral oil, glucose, and sucrose were obtained from Sigma Aldrich. Silica microparticles with a diameter of 2.12 μm were procured from Microparticles GmbH. We synthesized polystyrene particles that were sterically stabilized with poly-vinylpyrrolidone (molecular weight $M_{wt}$ = 40,000 kg/mol) and fluorescently labeled with rhodamine isothiocyanate, using the method of Song et al.[67]. After the synthesis, the particles were washed several times and re-dispersed into Milli-Q water. The particle size was 2.0 μm with a size polydispersity of 3%, as determined using static light scattering and scanning electron microscopy. A solution containing 10 wt.-% polyacrylamide (PAM) (MW 700,000 - 1,000,000) in water was obtained from Polysciences Inc.

**Vesicle preparation protocol.** Lipid oil solution (LOS) was prepared using either paraffin oil or mineral oil, following the adapted protocol by Vutukuri et al.[32]. In both cases, 1,2-dioleoyl-sn-glycero-3-phosphocholine (DOPC) and fluorescent 1,2-dioleoyl-sn-glycero-3-phosphoethanolamine-N-(lissamine rhodamine B sulfonyl) (ammonium salt) (Liss Rhod PE) were diluted in chloroform to final concentrations of 12 mg/mL (15 mM) and 0.2 mg/mL (0.15 mM), respectively, and stored at − 20 °C until use.

To prepare the LOS in paraffin oil, 0.31 g of DOPC and 0.12 g of Liss Rhod PE stock solutions were added to a clean 20 mL glass vial (Sample Storage Assembled Screw Vial Kits, Thermo Scientific). The chloroform was then evaporated under gentle $N_2$ airflow while rotating the vial to create an even layer of dried lipids on the bottom. The vial was placed in a desiccator for 2 hours to remove any remaining traces of chloroform. Subsequently, 2.2 g of paraffin oil was added to the vial, followed by sonication for 1 hour while heating the bath to 60 °C to enhance lipid solubilization. The LOS was then kept in a 60 °C oven overnight to ensure complete dissolution of the lipids and later stored at the same temperature.

The same steps were followed with a few modifications to prepare the LOS in mineral oil. In a clean glass vial, 0.2 g of DOPC and 0.08 g of Liss Rhod PE stock solutions were added. After drying and desiccation, 4 g of mineral oil was added to the vial. This was followed by 1 hour of sonication while heating the bath to 40 °C. Finally, the LOS in mineral oil was kept overnight at room temperature in the dark and later stored at 4 °C[68]. We note that the vesicles produced using paraffin and mineral oil did not show any difference. To prepare vesicles with silica particles inside, the droplet transfer protocol adapted from Vutukuri et al. was used[32]. The inner solution was composed of 2.12 $\mu$m silica particles (-0.125-0.25 wt.-%) in 100 mM sucrose solution, while the outer solution consisted of 110 mM glucose.

In a 2 mL Eppendorf tube, 200 $\mu$L of LOS was layered on top of 500 $\mu$L of the outer solution. In a separate 2 mL Eppendorf tube, 600 $\mu$L of LOS was mixed with 100 $\mu$L of the inner solution for 2-3 minutes using a 1 mL pipette to create an emulsion. Next, 120 $\mu$L of the emulsion, taken from the top of the second tube, was added to the water-oil interface in the first tube. The mixture was then immediately centrifuged (Centrifuge 5425, Eppendorf) at 200 g for 2 minutes, thus forming vesicles. Using a pipette, the top oil layer was carefully removed, leaving the vesicle solution at the bottom of the tube. The tube was left undisturbed for 30-60 minutes to allow the vesicles to accumulate before transferring them to the imaging chamber.

**Imaging.** The measurements were conducted using a confocal laser scanning microscope (Nikon eclipse Ti-U inverted microscope with a VTinfinity3 CLSM module, Visitech) equipped with a Hamamatsu ORCA-Flash4.0 CMOS camera and an oil objective lens (100x, 1.49 NA), and an inverted fluorescence microscope (Nikon eclipse TE2000-U) equipped with a Basler acA4112-30um CMOS camera and an oil objective lens (60x, 1.4 NA). To prepare for imaging, 20 $\mu$L of the vesicle solution was transferred to an 8-well chamber slide ($\mu$-Slide 8 Well Glass Bottom, Ibidi) and allowed to settle for several minutes. Occasionally, we observed that the vesicles were insufficiently deformed; in such cases, the well was left open for approximately 30 minutes, allowing the solvent to partially evaporate to further deflate the vesicles[69].

In some cases, to minimize the vesicle drift during scanning, a small amount of non-adsorbing polymer was added to the solution in the well, thus inducing depletion attractions between the vesicles and the bottom chamber wall. Typically, 12 $\mu$L of 0.2 wt.-% PAM in 110 mM glucose was gently added to the vesicle solution in the well, resulting in a final concentration of 0.075 wt.-% PAM. Note that we did not see any effect of the immobilization of vesicles on particle packing.

XY-Z scans were conducted in both composite (bright-field + fluorescence) and fluorescence mode to capture the vesicles' shape. The composite mode was used to visualize the location and arrangement of the particles and was generally conducted at a step size of 0.5 $\mu$m. The fluorescence mode was used to accurately determine the vesicle shape and was typically performed at a step size of 0.1 $\mu$m. Temporal recordings were taken in composite mode at 2–5 frames per second.

The osmotic imbalance between the inner and outer solutions resulted in the formation of flaccid vesicles[34,70]. We often observed flaccid vesicles with different shapes, such as ellipsoids and tubes containing a different number of particles. Fig. 2 depicts microscope images demonstrating the experimental state diagram of vesicles containing 3 to 9 particles exhibiting sausage, plate, and cluster-like configurations. Note that we also used sterically stabilized polystyrene particles to explore whether the experimental results were specific to the particle type or whether gravity had any impact. Our findings revealed that the observed trends were comparable to those obtained with the other particles used in the experiments. The observed particle arrangements were stable throughout the measurement time scale (≥20 minutes), and we did not observe any vesicle shape changes during the measurement. However, we note that to prevent shape changes of vesicles during imaging caused by laser heating[71], the laser intensity was minimized while searching for suitable vesicles. XY-Z scans were conducted first to ensure an accurate and undisturbed vesicle shape before temporal recordings. Lower laser intensity was used during temporal recordings to minimize potential heating effects.

**Analysis.** We extracted the bending rigidity and the tension of the vesicles using flickering spectroscopy[32,35]. The bending rigidity of our vesicles was $\kappa_c = 18 \pm 6\ k_BT$ and the membrane tension was $\lambda_p = 13 \pm 7$ nN m$^{-1}$.

The following steps were implemented to determine the volume and surface area of the vesicle using Fiji (ImageJ)[72]:

- The XY-Z-stack was imported into Fiji, followed by smoothing with a Gaussian filter and then binarizing using a threshold.
- Various functions (i.e., fill holes and despeckle) were applied to obtain a binary image where the vesicle and its interior are white, and the exterior of the vesicle is black.
- The Shape Smoothing plugin (v1.2) was applied to smooth the contour of the vesicle, followed by 3D Gaussian smoothing. The image was subsequently binarized again using a threshold.
- Finally, the binary stack was analyzed using the Particle Analyser function in the BoneJ plugin (v7.0.14,[73,74]) to obtain the vesicle volume (pixel count multiplied by voxel volume) and the vesicle surface area (surface mesh).

To correct for spherical aberration resulting from imaging vesicles in an aqueous medium with an oil objective lens, a correction factor was applied to the z-spacing. The correction factor was calculated using the ImageJ plugin described in ref. 75, with a numerical aperture of NA = 1.49, a refractive index of 1.52 for the immersion oil, and a refractive index of 1.33 for the imaging medium. The calculated correction factor was 0.83, which was validated using spherical fluorescent particles.

We noticed an artifact in the detection of the membrane contour, which was caused by slow Z-scanning (-3 fps) and the presence of membrane undulations. We corrected this artifact by selecting an appropriate 3D smoothing factor and estimated the associated error. The estimated error in the reduced volume for the cluster, plate, and linear configurations are 3%, 5%, and 8%, respectively.

## Numerical section

**Interaction potentials for the vesicle.** We employ the meshless model for vesicles presented in ref. 43. The particles that constitute the vesicle interact through an orientation-dependent potential, they have a unitary mass $m$, and diameter $\sigma$, which we take as the unit of length in our simulations. The interaction potential reads[43]

$$U(\mathbf{r}_{ij}, \mathbf{n}_i, \mathbf{n}_j) =$$
$$\begin{cases} U_R(r) + [1 - \phi(\mathbf{r}_{ij}, \mathbf{n}_i, \mathbf{n}_j)] & r < r_{min} \\ U_A(r)\phi(\mathbf{r}_{ij}, \mathbf{n}_i, \mathbf{n}_j) & r_{min} < r < r_c, \end{cases} \quad (2)$$

**Table 1 | Simulation parameters used in the meshless vesicle model**

| Parameter | Value | Parameter | Value |
|---|---|---|---|
| $\epsilon$ | 1.0 | $\xi$ | 4.0 |
| $\sigma$ | 1.0 | $\mu$ | 3.0 |
| $r_{min}$ | $2^{1/6}\sigma$ | $\sin\theta_0$ | 0 |
| $r_c$ | $2.6\sigma$ | $k_b T/\epsilon$ | 0.23 |

where $\mathbf{r}_{ij}$ is the distance vector between particle $i$ and $j$, $\mathbf{n}_{i(j)}$ is the unit vector that denotes the orientation of particle $i(j)$, $r_{min}$ is the distance that minimizes the attractive potential $U_A$, $U_R$ is a repulsive potential, and $\phi(\mathbf{r}_{ij}, \mathbf{n}_i, \mathbf{n}_j)$ an orientation function. We take

$$U_R(r) = \epsilon\left[\left(\frac{r_{min}}{r}\right)^4 - 2\left(\frac{r_{min}}{r}\right)^2\right]; \qquad (3)$$

$$U_A(r) = -\epsilon\cos^{2\xi}\left(\frac{\pi}{2}\frac{r - r_{min}}{r_c - r_{min}}\right), \qquad (4)$$

as the repulsive and attractive potential, respectively, where $\epsilon$ sets the energy scale, $\xi$ acts on the slope of the attractive potential and $r_c$ is the cutoff radius. The corresponding orientation dependent function $\phi(\mathbf{r}_{ij}, \mathbf{n}_i, \mathbf{n}_j)$ reads

$$\phi = 1 + \mu(a(\mathbf{r}_{ij}, \mathbf{n}_i, \mathbf{n}_j) - 1), \qquad (5)$$

where $\mu$ is a parameter related to the bending rigidity $\kappa_c$, and $a$ reads

$$a = (\mathbf{n}_i \times \mathbf{r}_{ij}) \cdot (\mathbf{n}_j \times \mathbf{r}_{ij}) + \sin\theta_0(\mathbf{n}_i - \mathbf{n}_j) \cdot \mathbf{r}_{ij} - \sin^2\theta_0, \qquad (6)$$

with $\theta_0$ a parameter related to the spontaneous curvature of the membrane. The aforementioned potential is already implemented in the LAMMPS simulation package[44]. Following ref. [43] we use the parameters shown in Table 1, which roughly correspond to a fluid vesicle with a bending rigidity $\kappa_c \approx 20 k_B T$ and an area compression modulus $K_A \approx 18 k_B T/\sigma^{2}$[42]. Since we are mainly interested in the volume-to-surface area ratio of the vesicles, the use of other bending parameters would yield the same phase diagram as shown in Fig. 1.

**Interaction potential for the colloids**

In all simulations, the colloidal spheres in the vesicle have a diameter of $\sigma_c = 12\sigma$ and interact via a soft repulsive Weeks-Chandler-Andersen (WCA) potential:

$$U_{\text{WCA}}(r) = \begin{cases} 4\epsilon\left[\left(\frac{\sigma_c}{r}\right)^{12} - \left(\frac{\sigma_c}{r}\right)^6\right] + \epsilon & \text{if } r \leq 2^{\frac{1}{6}}\sigma_c \\ 0 & \text{otherwise,} \end{cases} \qquad (7)$$

where $\epsilon$ sets the energy scale and $r$ is the distance between two colloids. We also use a WCA potential for the interaction between the colloids and the vesicle.

**Simulation protocol**

To obtain the state points shown in Fig. 1a, we modify the shape of the vesicle using solvent particles. The solvent is introduced outside the vesicle and exerts a pressure on the membrane. The interactions between solvent particles and solvent-vesicle are described in detail in ref. [43]. In order to generate vesicles with a low reduced volume $v$, we introduce solvent particles to the outer region of a pre-equilibrated membrane, to which $N$ colloidal particles are added. In this way, linear conformations can be observed. Subsequently, the solvent is gradually removed to achieve vesicles with higher $v$. By controlling the amount of solvent removal, we thus observe different configurations such as

planar, and cluster arrangements (see Fig. S2). Table S1 provides an overview of the key parameters, including the size of the membrane in its initial spherical state and the range of solvent particle densities used to explore various conformations.

## Data availability

Source data files are available with the paper. Source data are provided with this paper.

## Code availability

Simulations were performed with an open-source package as referenced in the manuscript. Analysis codes can be made available from the authors upon request.

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

## Acknowledgements

S.M.-A., F.C., and M.D. thank Gerardo Campos-Villalobos and Rodolfo Subert for early contributions to this work and for useful discussions. H.R.V. and S.v.d.H. thank Frieder Mugele and Mireille Claessens for kindly providing access to confocal and fluorescence microscopes.

## Author contributions

These authors contributed equally: S.M.-A., F.C., and S.v.d.H. H.R.V. and M.D. conceived and designed the project. S.M.-A. and F.C. performed the numerical simulations and analysis. S.v.d.H. and A.F. performed the experiments. S.v.d.H. performed the experimental analysis. H.R.V. supervised S.v.d.H. and A.F.. S.M.-A., F.C., S.v.d.H., H.R.V., and M.D. wrote the original draft of the paper. All authors participated in the discussions, and reviewed and edited the manuscript.

## Funding

S.M.-A., F.C. and M.D. acknowledge funding from the European Research Council (ERC) under the European Union's Horizon 2020 research and innovation program (Grant agreement No. ERC-2019-ADG 884902, SoftML). S.M.-A. and M.D. acknowledge funding from the Netherlands Center for Multiscale Catalytic Energy Conversion (MCEC). F.C. and M.D. acknowledge funding from the World Premiere International (WPI) Research Center Initiative of the Japanese Ministry of Education, Culture, Sports, Science and Technology (MEXT).

## Competing interests

The authors declare no competing interests.
