## [Peer Review File · Nature Communications]

REVIEWER COMMENTS

Reviewer #1 (Remarks to the Author):

=====

Report on MS. NCOMMS-23-33749

Authors: Marin-Aguilar et al

Title: A colloidal viewpoint on the finite sphere packing problem: the sausage catastrophe

=====

This manuscript reports on a combined experimental-numerical study on a problem originating from a series of mathematical papers whose basic idea is the following.

For infinite (or unconstrained) systems, Kepler conjectured and Hales proved that the optimal way of packing spheres is the FCC/HCP arrangement with 74% packing fraction. But what if the system is confined into a container with flexible and adaptive shape?

The authors discuss under which conditions a linear arrangement is favored against a cluster packing and provide numerical and experimental supports to their findings.

The manuscript is professionally written, the work is very elegant and instructive (especially the possibility of observing their prediction alluded toward the end of the manuscript is very fascinating) and it definitely deserves publication. However, there are few points that deserve the authors' attention because they puzzled me and I feel that they may hamper the full appreciation of this work.

1. The authors were stimulated by the mathematical results reported in Refs. 13-17 that are wrapped up in the introduction. However, it is not clear to me when the reported statements are proofs and when are conjectures. For instance when the authors say (page 1 first column) " On the contrary, in 1975 the mathematician Fejes-Toth theorized that the densest packing is always the linear arrangement..." what do they mean for "theorized"? Is that a proof or a conjecture requiring supporting evidence such as those provided in this study? As this literature is likely unfamiliar to most readers, it seems to me that a clarification of this point is desirable.

2. I don't understand the difference between a sausage and a three-dimensional cluster. Please explain. Also, I found confusing the reference to "finite" as opposed to infinite packing in the introductory statement. A finite systems with PBC would not behave as reported in the present study. As clarified later on in the manuscript, the crucial difference is between packing "constrained" into a membrane with zero surface tension so that the membrane can shape adapt to achieve the densest packing.

3. Is the orientation-dependent interaction Eq.(2) mimicking the floppy vesicle used in the experiment? I am not sure I can rationalize why. Please explain.

4. What is the difference between η and η_{ch} ? And how are the values $\eta \approx 0.12-0.28$ referred to Fig.S2 (page 2 second column) obtained?

5. As I gather from the introduction, the "sausage catastrophe" occurs in $d=4$ -- again not obvious whether a proof or a conjecture. Why the authors see their results in $d=3$ as "contradicting" with those of Refs. 15 and 16 (page 2 first column)?

6. The experimental results reported in this study build on the techniques from Refs. 9 and 23. Likewise, the numerical results hinge on those from Ref. 29 and 30. The authors duly report references to these study but it is not clear to me whether the present study also reports new insights on the techniques themselves or not.

Reviewer #2 (Remarks to the Author):

The manuscript by Marin-Aguilar et al. describes a combined experimental/simulation investigation into dense packing arrangements of finite numbers of spheres. As the authors point out, sphere packing has been of interest in a growing number of fields over the past few centuries. This manuscript presents a novel physical instance of the finite sphere packing problem via colloidal hard spheres enclosed in a lipid vesicle.

To my knowledge, the authors report a novel contribution to the literature, and the manuscript lays out what seems to be a well-executed and well-described investigation.

I have no quarrel with the quality of the work done in the manuscript, but I wonder if the authors could do a better job of situating its importance.

It seems like one of the main advances of the manuscript is to construct a physical realization of the problem, however it seems like the authors are only able to reproducibly generate vesicles with small numbers of colloids inside, whereas it seems like where non-sausage clusters, the structure of which less is known about, are inaccessible. On the other hand, the authors are able to address this question with their numerical simulations, the results of which come through clearly. However, if I am reading their conclusions correctly, it seems like the structure of larger dense clusters is not achievable inside flexible vesicles.

This latter effect, which the authors skim over somewhat, might appear to be a "bug". However, it could also be seen as a "feature" in the context of recent work showing that clusters of colloids with unconventional packing arrangements can be used as building blocks for hierarchically structured materials (e.g. Baldauf et al., *Sci Adv* 2022). Thus the fact that the authors produce could potentially produce packings other than the densest ones could provide leverage.

Perhaps I'm misreading the manuscript, but to me it seems like, given the framing in the introduction, the main contribution of the manuscript is to give a very detailed set of numerically constructed proposals for densely packed collections of spheres, particularly in the range of N between 50 and 75. That to me seems like an interesting result, and, given the longstanding interest in this problem and the array of applications of the result makes the paper one that should be interesting to readers of *Nature Communications*.

Overall, my assessment is that this is a valuable contribution, but the authors should more clearly articulate its value, particularly in final concluding paragraph.

As a side note, I think that it would further add value to the contribution to make more substantive connection to areas where the experimental setup the authors describe here could be leveraged. In addition to the example mentioned above on hierarchical assembly, there has also been work on colloidal clusters for wet computing (e.g. Phillips et al., *Soft Matter* 2014), among other applications, and the numerically constructed clusters reported here should be of interest to the math community.

Reviewer #3 (Remarks to the Author):

Review on "A colloidal viewpoint... the sausage catastrophe"

by Susana Marín-Aguilar et al.

The paper starts out with a motivation from the finite sphere packing problem, including works from Wills/Tóth and coworkers. It then shifts focus towards collids confined by "GUVs" of which the packing behavior is accessible by confocal microscopy. Systems up to $N=9$ enclosed colloid particles are studied away from the close-packing limit. Linear, flat plates and spheroidal clusters are found, depending on an effective surface/volume ratio. In addition to the experiments, MD simulations with WCA particles in a meshless vesicle are performed, the two methods appear to produce comparable results. Then, an optimization scheme is used to find dense arrangements of colloids. None surpass the 1D string. Subsequently, cuts of the FCC lattice are studied which yield the expected crossover close to ~ 60 spheres. Excluding $N=63,57$, the densest arrangements with $N \geq 56$ are found to be clusters.

The manuscript is well-written and it is clear what was done and why it was done.

The link between the confined colloid experiments and simulations and the infinite-pressure packing problem is a bit tenuous - it almost seems like there are two papers here in one manuscript. I wonder if the authors have a way, in

experiment, to control the surface/volume ratio of the cluster, for example by evaporation, and drive the system closer to dense packing.

The fact that a finite system oscillates between two conformations (plate/linear in Fig 1j) is rather unsurprising in my opinion. It is natural for clear cut 'phase transitions' to emerge only in the large-N limit. Can the authors comment why they find this surprising?

It would also be extremely interesting why the contradiction to Ref. [15,16] with regards to $N=58$ and $N=64$ can be resolved. This is a mathematical problem which should have a robust answer.

Questions/suggestions:

Is the plate-like phase (Fig 1a) expected to disappear in the large-N limit?

Can the authors comment in the relative importance of the conformation entropy of the vesicle and the colloids?

Were other Barlow stacking types than FCC considered (for example HCP) to explore the densest packings? Since the problem is so subtle, stacking order could make a difference.

A Monte Carlo scheme adding/removing spheres at the boundary of the cluster might be able to find other candidates for the densest packing.

Are the findings with respect to closest packing stable with respect to the definition of packing fraction (here via cluster convex hull)?

REPLY TO THE COMMENTS OF REVIEWER 1

This manuscript reports on a combined experimental-numerical study on a problem originating from a series of mathematical papers whose basic idea is the following. For infinite (or unconstrained) systems, Kepler conjectured and Hales proved that the optimal way of packing spheres is the FCC/HCP arrangement with 74% packing fraction. But what if the system is confined into a container with flexible and adaptive shape? The authors discuss under which conditions a linear arrangement is favored against a cluster packing and provide numerical and experimental supports to their findings. The manuscript is professionally written, the work is very elegant and instructive (especially the possibility of observing their prediction alluded toward the end of the manuscript is very fascinating) and it definitely deserves publication. However, there are few points that deserve the authors' attention because they puzzled me and I feel that they may hamper the full appreciation of this work.

We thank the Reviewer for finding our manuscript instructive and well-written and for suggesting publication after having clarified a few aspects.

1. The authors were stimulated by the mathematical results reported in Refs. 13-17 that are wrapped up in the introduction. However, it is not clear to me when the reported statements are proofs and when are conjectures. For instance when the authors say (page 1 first column) "On the contrary, in 1975 the mathematician Fejes-Tóth theorized that the densest packing is always the linear arrangement..." what do they mean for "theorized"? Is that a proof or a conjecture requiring supporting evidence such as those provided in this study? As this literature is likely unfamiliar to most readers, it seems to me that a clarification of this point is desirable.

We thank the Reviewer for pointing out this aspect, which helped us to clarify the introduction and the context of our manuscript. In summary, the finite sphere problem was initiated by Fejes Tóth in 1975 [Fejes Tóth, Research problem no. 13, Period. Math. Hungar. (1975)] who conjectured that, in dimension $d \geq 5$, the highest packing of non-overlapping spheres is that given by their linear arrangement. This conjecture was further supported by later works such as [Betke, Gritzmann and Wills, Mathematika (1982); Fejes Tóth, Gritzmann and Wills, Disc. Comp. (1989)]. Proof of the conjecture was first given in 1994 for $d \geq 13387$ [Betke, Henk, Wills, J. Reine. Angew. Math. (1994)] and later for $d \geq 42$ [Betke and Henk, Discrete Comput. Geom. (1998)], as already mentioned in the manuscript. Therefore, there exists no formal proof of the validity of the conjecture for lower dimensions. In the context of $d = 3$, several configurations have been analyzed mathematically in the past [Wills, Acta Mathematica Hungarica (1985); Wills, Periodica Mathematica Hungarica (1983); Gandini and Wills, Math. Pannon. (1992)]. However, as demonstrated in our work, it remains uncertain under which

conditions linear conformations constitute the densest packings (see also our response to a later point) or when a different type of cluster could surpass linear packing. As the Reviewer correctly states, we indeed provide supporting evidence to the original conjecture.

We have now revised the introduction and included additional references to provide further clarification on the state of the finite sphere packing problem.

We revised the Introduction and included additional references.

Introduction:

On the contrary, in 1975 the mathematician Fejes Tóth conjectured [14] that in dimensions $d \geq 5$ the densest packing is the one where the centers of spheres are aligned along a straight line, resulting in a so-called *sausage configuration*. This conjecture, supported by other studies [15, 16], was initially proven true only for $d \geq 13387$ [17] and subsequently for $d \geq 42$ [18]. No proof of its general validity has thus been reported for lower dimensions. For $d = 4$, a sudden transition in the packing density occurs from a linear to a cluster arrangement, where the coordinates of the particles extend in all three dimensions, and is typically referred to as the “sausage catastrophe” [19, 20]. The upper bound of $N = 375769$ that was initially assigned to this transition [21, 22] has been recently reduced to $N = 338196$ [23]. For $d = 3$, different studies reported that the sausage conformation minimizes the volume of the convex hull for $N \leq 55$, and for $N = 57, 58, 63, 64$ [19, 24, 25], while above this limit the densest configuration becomes a three-dimensional cluster, thereby avoiding the plate conformation where the centers of the spheres are positioned on a plane [15, 26]. However, the precise structure of these clusters, which are denser than the sausage, remains largely unknown. Furthermore, there may be other unidentified clusters that present an even denser packing than the linear arrangement.

2. I don't understand the difference between a sausage and a three-dimensional cluster. Please explain. Also, I found confusing the reference to "finite" as opposed to infinite packing in the introductory statement. A finite systems with PBC would not behave as reported in the present study. As clarified later on in the manuscript, the crucial difference is between packing "constrained" into a membrane with zero surface tension so that the membrane can shape adapt to achieve the densest packing.

We appreciate the Reviewer's question that allows us to better explain the terminology used in our manuscript. In a 'sausage' configuration, particles are aligned linearly, essentially forming a one-dimensional arrangement, i.e. the center-of-masses of the particles are positioned on a one-dimensional line. In contrast, a 'cluster' refers to a collection of particles that are closely grouped together in a three-dimensional arrangement. We illustrate both configurations in Figure 2 of the main text, providing examples from simulations and experiments.

The term “finite packing” is employed to describe the arrangement of a limited number of particles within a defined space, while an “infinite packing” is an arrangement of particles that extends to infinity in all directions. To calculate the density of a finite packing of spheres, one has to define the volume that the spheres occupy by specifying the spatial boundaries that encloses the spheres. The distinction between “finite” and “infinite” packing is thus not based on whether the system employs periodic boundary conditions (PBC) but rather on the spatial constraints imposed on the particles.

In a system with PBC but no spatial constraints, the particles behave as if they are part of an infinite system, similar to crystalline or percolating systems that are often simulated with PBC [see, for instance, Griffiths, Turci, and Royall, *J. Chem. Phys.* (2017); Marín-Aguilar, Camerin and Dijkstra, *J. Chem. Phys.* (2022)]. In contrast, in our study, a ‘finite’ system is constrained by its boundaries, which can either be rigid or flexible.

These definitions have been now clarified in the revised version of the manuscript.

These definitions have been clarified in the revised Introduction.

Introduction:

In 1611, Kepler conjectured that the densest packing of an *infinite* number of identical, non-overlapping spheres in three-dimensional (bulk) conditions is the “cannonball” stacking or the face-centered cubic (FCC) crystal [...] For $d = 4$, a sudden transition in the packing density occurs from a linear to a cluster arrangement, where the coordinates of the particles extend in all three dimensions.

3. Is the orientation-dependent interaction Eq.(2) mimicking the floppy vesicle used in the experiment? I am not sure I can rationalize why. Please explain.

We appreciate the Reviewer’s suggestion to clarify this. Indeed, the orientation-dependent interaction employed in the model for flexible vesicles accounts for the nature of its lipid constituents, encompassing the various tail-tail, head-head and head-tail (hydrophilic and hydrophobic) interactions that enable real GUVs to maintain their closed shape [Fu et al. *Computer Physics Communications* (2017)]. Without taking into account these interactions, the vesicle model would behave differently. These details have now been incorporated in the revised version of our manuscript to enhance clarity and accuracy.

A comment regarding the rationale for incorporating an orientation-dependent interaction has now been included in the description of the model.

Results and discussion:

In the molecular dynamics simulations, various vesicle shapes are investigated using a meshless membrane model [42, 43]. In this model, lipids are represented in a coarse-grained fashion using spheres of diameter σ , which is also used as the unit of length in our simulations. The model incorporates orientation-dependent interactions to account for the properties and interactions of the constituent lipids in real GUVs.

4. *What is the difference between η and η_{ch} ? And how are the values $\eta \approx 0.12 - 0.28$ referred to Fig.S2 (page 2 second column) obtained?*

We thank the Reviewer for giving us the opportunity to elucidate this in the revised manuscript. The quantity $\eta = NV_0/V_v$, where N represents the number of colloids, V_0 is the volume of a single colloid, and V_v is the volume of the vesicle, serves as the packing fraction for colloids within the flexible vesicle. This is determined by utilizing the surface mesh of the vesicle. From this estimation, values of η in the range of approximately 0.12 to 0.28 are extracted. These values correspond to the highest and lowest packing fractions among the state points presented in Figure 1a and S1.

Conversely, the packing fraction of colloids enclosed within a convex hull is calculated as $\eta_{ch} = NV_0/V_{ch}$, where V_{ch} denotes the volume of the convex hull, computed using the method described in the Supplementary Information. As discussed in the text, using η_{ch} allows us to effectively study the packing fraction within the tightest possible container. This can be compared, for instance, with the ideal packing fraction of a linear arrangement of colloids, which is the one given by the spherocylinder. The definition of η has been now better specified both in the main text and in the Supplementary Information.

We have added the definition of η in the Results and discussion section, and included a paragraph on the estimate of V_v via the surface mesh in the Supplementary Information.

Results and discussion:

We observe that state points obtained from simulations and experiments encompass a wide range of packing fractions $\eta = NV_0/V_v \approx 0.12 - 0.28$ for the colloids in the vesicle (see Fig. S1), with V_0 representing the volume of a colloid and V_v the volume of the vesicle. The latter is estimated by generating the surface mesh of the vesicle [45, 46], which allows us to directly extract the value of the volume enclosed within the meshless membrane (see SI).

Supplementary Information:

For the calculation of the packing fraction η , the volume of the vesicle is estimated by means of its surface mesh with the alpha-shape method of the OVITO software, which provides a three-dimensional surface representation of the vesicle [1, 2]. This method relies on a Delaunay

tessellation constructed on the basis of the input particle coordinates and uses a probe sphere with prescribed radius $R_{probe} = 10 - 16\sigma$ (depending on the size of the vesicle) to assign each tetrahedral element to a region of space, of which the volume is calculated.

5. *As I gather from the introduction, the "sausage catastrophe" occurs in $d=4$ – again not obvious whether a proof or a conjecture. Why the authors see their results in $d=3$ as "contradicting" with those of Refs. 15 and 16 (page 2 first column)?*

We appreciate the Reviewer's comment. In fact, the term "sausage catastrophe" has been used for dimensions other than $d = 4$ as well. For example, it was used in the context of $d = 3$ with $N = 56$ to indicate when the linear conformation ceases to have the optimal packing [see Gandini and Wills, Math. Pann. (1992)].

Regarding the perceived contradiction, we acknowledge that using clearer wording in our manuscript would have better conveyed our intended meaning. We apologize for any misunderstanding. In fact, our work, does not contradict previous results but rather uncovers clusters that were not previously identified or explicitly examined in [Gandini and Wills, Math. Phann. (1992)]. Whether a rigorous mathematical proof can confirm that these clusters have a higher packing remains an open question, and goes beyond the scope of this paper. We believe that our work will spark the interest of an interdisciplinary community and may eventually find formal mathematical validation. We have now rephrased the corresponding part of the manuscript.

The final paragraph of the introduction has been rewritten.

Introduction:

Finally, we identify the conditions required to form finite clusters with high packing efficiency for a large number of spheres and study them systematically. In this way, we uncover clusters composed of $N = 58$ and 64 spheres that exhibit better packing than the linear conformation. As a result, we provide evidence for the existence of particle arrangements with higher packing efficiency compared to those previously examined [19, 25], thereby lending direct support to Fejes Tóth original conjecture.

6. *The experimental results reported in this study build on the techniques from Refs. 9 and 23. Likewise, the numerical results hinge on those from Ref. 29 and 30. The authors duly report references to these study but it is not clear to me whether the present study also reports new insights on the techniques themselves or not.*

We thank the Reviewer for their comment. While we did not introduce new simulation techniques beyond those reported in [Fu et al., Computer Physics Communications (2017) (Ref. 30

in the original manuscript)], our approach distinguishes itself through a systematic investigation of controlling the vesicle shape and particle packing within a flexible container. Specifically, we modified the vesicle shape and size by manipulating the number of solvent particles in its outer region. This is in contrast to the approach taken in Ref. 30, where adjustments were made by varying the size of the inner solvent particles.

Regarding our experimental method, we encapsulated particles within giant unilamellar vesicles (GUVs) using a slightly modified droplet transfer method, drawing inspiration from the technique employed in [Vutukuri et al. Nature, (2020) (Ref. 23 in the original manuscript)]. While our method bears some similarities, it is essential to emphasize that the primary focus of our study differs substantially. In Ref. 23, active or self-propelled particles were used to study how localized forces deform the lipid membrane, resulting in non-equilibrium membrane fluctuations. In contrast, the present study focuses solely on the packing behavior of ‘passive particles’ in GUVs with very low interfacial tension.

Incidentally, we wish to clarify that our method is not based on the work described in [Manoharan et al. Science, (2003) (Ref. 9 in the original manuscript)]. In that study, particles were encapsulated in oil emulsion droplets stabilized by surfactants in water. The resulting tension of these droplets is several orders of magnitude higher than that of GUVs. Therefore, we can consider them as ‘hard containers’, where it is very difficult to vary their droplet tension. In contrast, in our work the tension in GUVs can be precisely controlled across a wide range, from mN/m to nN/m, using osmotic shock. Therefore, we consider them as ‘flexible containers’.

For more details on this, please refer to our response to question 1 from Reviewer 3, where we experimentally demonstrate how changes in the surface-to-volume ratio of the vesicle can lead to a transition between linear and clustered arrangements of particles, and vice versa.

We have added this information in the revised manuscript.

Results and discussion:

In experiments, we use a modified droplet transfer method [34] to encapsulate colloidal particles of size $2.12 \mu\text{m}$ in GUVs, drawing inspiration from a previous work by Vutukuri et al. [32]. [...] We note that, while Ref. [32] dealt with self-propelled particles locally deforming the lipid membrane, here we entirely focus on passive particles. Furthermore, the vesicles employed in this study significantly differs from oil emulsion droplets [10], whose surface tension is several order of magnitude higher than that of GUVs. [...]

We use explicit solvent to control the shape of the vesicle but, differently from the original model, we only add it to the outer region thus exerting an external pressure on the membrane.

REPLY TO THE COMMENTS OF REVIEWER 2

The manuscript by Marin-Aguilar et al. describes a combined experimental/simulation investigation into dense packing arrangements of finite numbers of spheres. As the authors point out, sphere packing has been of interest in a growing number of fields over the past few centuries. This manuscript presents a novel physical instance of the finite sphere packing problem via colloidal hard spheres enclosed in a lipid vesicle. To my knowledge, the authors report a novel contribution to the literature, and the manuscript lays out what seems to be a well-executed and well-described investigation. I have no quarrel with the quality of the work done in the manuscript, but I wonder if the authors could do a better job of situating its importance.

We thank the Reviewer for their kind words on our work and for finding it sound and well-described in the manuscript.

1. It seems like one of the main advances of the manuscript is to construct a physical realization of the problem, however it seems like the authors are only able to reproducibly generate vesicles with small numbers of colloids inside, whereas it seems like where non-sausage clusters, the structure of which less is known about, are inaccessible. On the other hand, the authors are able to address this question with their numerical simulations, the results of which come through clearly. However, if I am reading their conclusions correctly, it seems like the structure of larger dense clusters is not achievable inside flexible vesicles. This latter effect, which the authors skim over somewhat, might appear to be a "bug". However, it could also be seen as a "feature" in the context of recent work showing that clusters of colloids with unconventional packing arrangements can be used as building blocks for hierarchically structured materials (e.g. Baldauf et al., Sci Adv 2022). Thus the fact that the authors produce could potentially produce packings other than the densest ones could provide leverage.

We appreciate the Reviewer's comments and for their suggestion to turn a bug into a feature. Indeed, it is rather challenging to realize both in simulations and in experiments systems containing a large number of particles. As mentioned in the text, the main problems concern the excessive bending we would observe for vesicles with $N > 9$, and the high computational cost associated with including explicit solvent in such a system. For these reasons, we approach the study of configurations with a high number of colloids differently in the second part of the manuscript.

We agree with the referee that the structures we observed in the vesicle have the potential to be used as preassembled building blocks for larger-scale exotic structures, as suggested in the reference provided by the Reviewer. We have also recently found other works where authors describe the use of small colloidal or nano-scaled clusters for applications as plasmonic meta-

molecules [see, for example, Huh et al., *Advanced Materials* (2020)]. We have incorporated these references and the Reviewer's suggestion into the main text of the manuscript.

We have revisited the concluding section of our manuscript and included reference to the above-mentioned papers.

Conclusions:

Beyond addressing the finite sphere packing problem, our findings have broader applications. For instance, the encapsulation of a limited number of spheres within a vesicle provides a strategy for pre-assembling building blocks that could be used to construct larger, more intricate structures [57, 58], with potential applications such as the enhancement of plasmonic properties in metamaterials [59]. Moreover, our methodology can be adapted to other building blocks like dimers, trimers, or tetramers, drawing inspiration from existing techniques that use patchy interactions or emulsion methods to realize these clusters [10, 60-62].

*2. Perhaps I'm misreading the manuscript, but to me it seems like, given the framing in the introduction, the main contribution of the manuscript is to give a very detailed set of numerically constructed proposals for densely packed collections of spheres, particularly in the range of N between 50 and 75. That to me seems like an interesting result, and, given the longstanding interest in this problem and the array of applications of the result makes the paper one that should be interesting to readers of *Nature Communications*. Overall, my assessment is that this is a valuable contribution, but the authors should more clearly articulate its value, particularly in final concluding paragraph.*

We appreciate the insights of the Reviewer regarding our work. In addition to providing a first physical realization of the finite sphere packing problem, our study indeed places a strong emphasis on analyzing in detail the range of particle numbers where the sausage catastrophe was expected to occur. To better emphasize this aspect, we have included a dedicated paragraph in the concluding discussion of the manuscript.

We have revised the concluding section of our manuscript accordingly.

Conclusions:

Our systematic investigation of these clusters with various shapes has allowed us to directly prove, with a practical approach inspired by the physics of colloids, the existence of previously unidentified clusters that exhibit a packing efficiency that is superior to the sausage configuration, highlighting that the finite sphere packing problem is still open and intriguing. Nevertheless, it remains to be determined whether mathematical proofs can be developed for the packing of these clusters and for the entire Fejes Tóth conjecture. We believe that our work can serve as a

catalyst for further research in this direction. From a computational perspective, we envision the development of cluster generation techniques, either through conventional or machine-learning methods [63]. Such approaches could expand the exploration of even more configurations and different Barlow stacking arrangements of spheres.

3. As a side note, I think that it would further add value to the contribution to make more substantive connection to areas where the experimental setup the authors describe here could be leveraged. In addition to the example mentioned above on hierarchical assembly, there has also been work on colloidal clusters for wet computing (e.g. Phillips et al., Soft Matter 2014), among other applications, and the numerically constructed clusters reported here should be of interest to the math community.

We thank the Reviewer for pointing out this additional example, which highlights another potential application of cluster formation. The reference to work on colloidal clusters for wet computing is now cited in our manuscript. Due to the rapid advances in simulation techniques, some of which have been developed in recent years, we have the opportunity to explore various ways to expand upon our current experimental systems. Specifically, we can consider employing different starting building blocks such as dimers, trimers, and even tetramers. These well-defined clusters could potentially be realized using various techniques, including emulsion droplet methods, as discussed in [Manoharan et al. Science, (2002)]. Another approach could involve patchy interactions, as described in several studies [see, for example Gong et al., Nature, (2017); He et al., Nature, (2020); Kim et al., JACS, (2021)]. By combining these methods and building blocks, we might be able to generate versatile structures with a wide range of functional properties.

We have revisited the concluding section of our manuscript and added the reference mentioned by the Reviewer in the Introduction.

Introduction:

Sphere packings also have applications in coding theory, wet computing, crystallography, and in understanding mechanical and geometrical properties of materials [2-7].

REPLY TO THE COMMENTS OF REVIEWER 3

The paper starts out with a motivation from the finite sphere packing problem, including works from Wills/Tóth and coworkers. It then shifts focus towards colloids confined by "GUVs" of which the packing behavior is accessible by confocal microscopy. Systems up to $N = 9$ enclosed colloid particles are studied away from the close-packing limit. Linear, flat plates and spheroidal clusters are found, depending on an effective surface/volume ratio. In addition to the experiments, MD simulations with WCA particles in a meshless vesicle are performed, the two methods appear to produce comparable results. Then, an optimization scheme is used to find dense arrangements of colloids. None surpass the 1D string. Subsequently, cuts of the FCC lattice are studied which yield the expected crossover close to ≈ 60 spheres. Excluding $N = 63, 57$, the densest arrangements with $N \geq 56$ are found to be clusters. The manuscript is well-written and it is clear what was done and why it was done.

We thank the Reviewer for the careful assessment of our manuscript and for considering it clear and well-justified.

1. The link between the confined colloid experiments and simulations and the infinite-pressure packing problem is a bit tenuous - it almost seems like there are two papers here in one manuscript.

We thank the Reviewer for their comment. We believe that the connection between the observation of different conformations in the vesicle and the analysis of higher- N clusters is manifold. On one hand, the number of colloids that can be encapsulated in a vesicle is limited. Therefore, the study of clusters with $N > 9$ allows us to determine which particle arrangements would result in a higher packing efficiency than the linear conformation in a region where the sausage catastrophe is expected to occur. For N up to 9, we can confirm that the packing is maximized for linear conformations through a physical realization of the finite sphere packing problem.

On the other hand, we have the opportunity to explore particle states at their highest packing, allowing us to compare them to the ideal packing fraction given by the convex hull. Experimentally, achieving states with very high packing fractions would be much more challenging because it would not be possible to directly manipulate the volume-to-surface area ratio to tightly enclose colloidal particles within vesicles. These aspects discussed in the manuscript are therefore conceived as complementary. We have made changes in the text to better emphasize this connection.

This reasoning has been introduced in the Results and discussion section.

Results and discussion:

After implementing an optimization protocol to approach the hard-sphere limit, we determine

the colloid packing fraction for both data sets as $\eta_{ch} = NV_0/V_{ch}$, where V_{ch} is the volume of the convex hull that encloses the colloids. This numerical protocol complements the analysis just presented on the flexible vesicle. In fact, besides being able to explore states with a larger number of colloids, by using η_{ch} we effectively study the packing fraction of the tightest possible container, and thus compare it to the ideal linear packing fraction η_{lin} obtained from the volume of a spherocylinder with N particles (see SI). In contrast, achieving states with a very high packing fraction experimentally would be significantly more challenging. This is because it is not feasible to directly manipulate the volume-to-surface area ratio to tightly enclose the colloidal particles within the vesicles.

2. I wonder if the authors have a way, in experiment, to control the surface/volume ratio of the cluster, for example by evaporation, and drive the system closer to dense packing.

We value the Reviewer's suggestion about the potential of controlling the surface/volume ratio of the vesicle to optimize cluster packing and find the analogy drawn with emulsion droplets to be highly relevant. Unlike emulsion droplets, simply evaporating the vesicle does not necessarily lead to maximum cluster packing due to the elasticity of vesicles, which allows them to transform into different shapes. The change in vesicle shape due to osmotic shock is attributed to a change in volume, while the surface area of the vesicle remains relatively constant. Consequently, preserving the shape of the vesicle while it deflates poses a considerable challenge.

However, following the Reviewer's suggestion, we performed additional experiments and simulations. In our experiments, we successfully manipulated the surface area-to-volume ratio of the vesicle through osmotic imbalances, enabling the observation of a transition from a linear arrangement to a clustered state within a single vesicle. Correspondingly, in simulations, a reduction in solvent density facilitated the transition of particles from a linear to a clustered conformation. The results from these supplementary experiments and simulations have been incorporated and are presented in Figure R1, Figure S3, and Movie S3.

We added a paragraph in the main text, and a section, figure and a video in the Supplementary Information.

Results and discussion:

To highlight the robustness of our methods, we convincingly demonstrate the transition of the particles from a linear arrangement to a clustered state both in simulations and experiments. This transition is achieved by precisely controlling the surface area-to-volume ratio of the vesicle through osmotic imbalances across the membrane, as depicted in Figure S3 and Movie S3.

Supplementary Information:

Figure R1: (a-d) Sequence of time-lapse images from 2D composite confocal and bright field microscopy, illustrating the transition from linear to cluster conformation as osmotic imbalances are induced in a vesicle containing 5 particles. (e-h) Corresponding simulation configurations generated by systematically changing the outer solvent density ρ_{sol} . (i) Dependence of the vesicle volume V_v on ρ_{sol} for the simulations shown in (d-f), with dashed lines serving as guides-to-the-eye to denote the transitions between the three different configurations.

To demonstrate the robustness of our experimental method, we show the transition of particles from a linear arrangement to a cluster state by precisely controlling the surface area-to-volume ratio of the vesicle. We induce transitions between the three distinct conformations by gradually increasing the vesicle volume while keeping the membrane surface area constant. We achieve this through a step-wise reduction in the osmolarity of the external solution, leading to a vesicle volume expansion. Particles in GUVs are initially arranged linearly, then they are subjected to an osmotic shock through the sequential addition of water outside the vesicle solution. In a typical experiment, we add a total of 50 μL of Milli-Q water in increments of 10 μL to the 40 μL vesicle solution. The first addition induces transition from the sausage to the plate conformation. The second addition transforms the plate to a cluster arrangement. Finally, the last three additions progressively transform the vesicle into a more spherical shape. We observe that each transition took place approximately 3-4 minutes following each addition. A similar procedure is done in simulations, where we gradually change the shape of the vesicle by removing solvent particles, thus changing the density of the solvent. This allows us to observe the transformation from cluster to linear conformation.

3. *The fact that a finite system oscillates between two conformations (plate/linear in Fig 1j) is rather unsurprising in my opinion. It is natural for clear cut 'phase transitions' to emerge only in*

the large- N limit. Can the authors comment why they find this surprising?

We thank the Reviewer for their comment. While it is indeed expected for the system to oscillate between plate-linear and plate-cluster states, bistable regions have never been identified before for particles enclosed in a flexible container. Most importantly, we provide the value of the parameter ν at which such transitions are expected to occur. We have now amended the text to better convey this message.

We have revised the corresponding sentence in the manuscript.

Results and discussion:

Additionally, we find bistable regions due to the combined membrane, shape, and solvent fluctuations driven by the colloids inside the vesicle.

4. It would also be extremely interesting why the contradiction to Ref. [15,16] with regards to $N=58$ and $N=64$ can be resolved. This is a mathematical problem which should have a robust answer.

We thank the Reviewer for pointing this out, and we apologize for any confusion. Our choice of words may not have adequately conveyed what we actually meant. Instead of contradicting the previous work cited in Refs. [15,16], our study unveils clusters that were not previously identified or explicitly discussed in those papers [Gandini and Wills, *Math. Phann.* (1992)]. While the question of formally proving the high packing of these newly discovered clusters remains an open topic, it lies beyond the scope of this paper. Nevertheless, we hope that our findings will spark interest across disciplines and may eventually lead to a formal mathematical validation. We have now rephrased the corresponding part of the manuscript.

The final paragraph of the Introduction has been rewritten.

Introduction:

Finally, we identify the conditions required to form finite clusters with high packing efficiency for a large number of spheres and study them systematically. In this way, we uncover clusters composed of $N = 58$ and 64 spheres that exhibit better packing than the linear conformation. As a result, we provide evidence for the existence of particle arrangements with higher packing efficiency compared to those previously examined [19, 25], thereby lending direct support to Fejes Tóth original conjecture.

5. Is the plate-like phase (Fig 1a) expected to disappear in the large- N limit?

The question raised by the Reviewer is indeed intriguing. Based on the phase diagram in Figure 1a, one might expect that planar conformations would remain stable even for a large number of particles. Additionally, as illustrated in Figure 4a, none of the planar arrangements achieves a packing efficiency that is higher than, or even close to, that of the sausage conformation, which is consistent with Ref. [15]

We added a comment on this in the Results and Discussion section.

Results and discussion:

Nonetheless, based on the state diagram presented in Figure 1(a), we expect that planar conformations would remain stable even for a large N .

6. Can the authors comment in the relative importance of the conformation entropy of the vesicle and the colloids?

We thank the Reviewer for raising this interesting point. Indeed, the transitions from linear to plate and cluster conformations may be affected by the configurational entropy of the vesicle and the colloids. However, we do not expect a great difference of configurational entropy among the states adopted by the colloids in the vesicle, considering a similar wiggling of the particles wrapped in a tight container. We can also expect an increase in configurational entropy for higher N . We agree with the Referee that it would be interesting to quantify the configurational entropy for the different conformations by taking into account the membrane and particle fluctuations, using for instance the simulations as presented here, or the approach presented in Ref. [13], or a cell-like theory where the shape of the vesicle and the particles are kept fixed, but we regard this to be outside the scope of our current study.

7. Were other Barlow stacking types than FCC considered (for example HCP) to explore the densest packings? Since the problem is so subtle, stacking order could make a difference.

We agree with the Reviewer that this is a very interesting point which indeed deserves further investigation. In fact, while FCC, HCP and other Barlow stackings have the same packing efficiency ($\approx 74\%$) in bulk, the different stacking of particle layers (A, B, C) might lead to differently packed structures in finite systems. For this reason, we generated a series of polyhedra starting from a bulk lattice with HCP (ABABAB) and two other orderings (ABABAC and ABACBC). For each polyhedron, we calculated its convex hull, and thus its volume and packing fraction, similar to the previously reported clusters. As summarized in Figure R2, we did not find any cluster with a better packing efficiency than the linear arrangement, even for irregularly cut polyhedra. It thus appears that the FCC ordering is best suited for generating clusters with the highest packing

Figure R2: Packing fraction η_{ch} of clusters of spheres in their convex hull as a function of N for the linear arrangement of spheres (orange line) and for (a) HCP-based (ABABAB stacking) pyramids and bipyramids, and for (b) irregularly cut HCP-based, ABABAC- and ABACBC-based polyhedra as compared to tetrahedra and octahedra based on an FCC stacking of the particle layers (dashed lines).

fraction when confined in the tightest possible container. We now included a comment on this in the main text and the most relevant graphs and snapshots in the Supplementary Information.

We added a paragraph in the main text, and a section, a figure and a table in the Supplementary Information.

Results and discussion:

In the Supplementary Information, we also present a similar analysis for other representative Barlow stacking arrangements of spheres [53-55], such as hexagonal close packing (HCP). Our findings demonstrate that these arrangements generally provide less efficient packings compared to the FCC.

Supplementary Information:

Along with the face-centered cubic (FCC) arrangement, it was proven that for an infinite number of spheres there exists an infinite number of other stackings that achieve the same packing efficiency of $\approx 74\%$ [6]. The latter are better known as Barlow stackings [7], and they are all based on variations of the three ways of accommodating a hexagonal layer on top of another. Each layer can be identified by the letters 'A', 'B' and 'C', with different combinations encoding for different stackings. The most regular and well known stackings are the hexagonal-closed packing (HCP) and the FCC arrangements with repeated sequences 'AB' and 'ABC' respectively. Here, we extend the study of the FCC-based structures presented in the main text by building clusters made of 6 layers with the 'ABABAB' sequence corresponding to HCP, 'ABABAC' (seq 1) and 'ABACBC'

(seq 2) [8]. In Fig. R2(a) we show the convex hull packing fraction η_{ch} of HCP-based pyramid (P_{HCP}) and bipyramid (B_{HCP}) clusters compared to the corresponding structures based on an FCC arrangement. We find that the different stacking introduced by the HCP arrangement is not as favorable as the one formed by the FCC for packing finite structures. A similar situation occurs with the other stackings, as shown in Fig. R2(b), where we show the η_{ch} of the pyramids based on the seq 1 and seq 2 (dashed symbols). Moreover, we perform irregular cuts to clusters based on HCP, seq 1 and seq 2, finding that all of them have a significantly lower η_{ch} compared to η_{lin} . We report in Table S4 the corresponding clusters with their convex-hull packing fraction η_{ch} when compared with the linear convex-hull packing fraction η_{lin} and the *3D View* which gives the short name of HTML files that contain the interactive three-dimensional visualization of the cluster.

8. A Monte Carlo scheme adding/removing spheres at the boundary of the cluster might be able to find other candidates for the densest packing.

We thank the Reviewer for this remark. Using a Monte Carlo-like scheme for finding new clusters is indeed a valuable suggestion. While the concept is straightforward, its implementation requires certain constraints and rules regarding the types of clusters that can be generated in order to limit their number and establish specific cuts on the polyhedra. Another promising approach for creating densely packed clusters involves the application of machine learning techniques, similar to what has been proposed, for example, in the work of Zang and Dolg, *Physical Chemistry Chemical Physics* (2015). We are actually pursuing this direction to compile a large pool of cluster candidates for an in-depth study and analysis. A comment on this aspect has been added to the conclusions of the manuscript.

We have revisited the concluding section of our manuscript and included reference to the above-mentioned paper.

Conclusions:

From a computational perspective, we envision the development of cluster generation techniques, either through conventional or machine-learning methods [63]. Such approaches could expand the exploration of even more configurations and different Barlow stacking arrangements of spheres.

9. Are the findings with respect to closest packing stable with respect to the definition of packing fraction (here via cluster convex hull)?

We thank the Reviewer for this comment. Indeed, the definition of the convex-hull packing

fraction $\eta_{ch} = NV_0/V_{ch}$, with V_0 the volume of each particle and V_{ch} the volume of the convex-hull (as described in the SI), is equivalent to the common packing fraction $\eta = NV_0/V_v$ for the tightest possible vesicle. To further confirm the accuracy of our approach, we compare the resulting η_{ch} with the estimated theoretical packing fractions for linear conformations η_{lin} made from 1 to 100 spheres. The latter is calculated as the packing fraction of a perfect spherocylinder as $\eta_{lin} = 2N/(2 + 3(N - 1))$. We report the difference between these two estimates as $\Delta\eta(N) = \eta_{lin} - \eta_{ch}$ in Figure R3, and we find that we have a precision up to the fourth decimal digit. This comparison is now also reported in the Supplementary Information.

Figure R3: Difference in packing fraction $\Delta\eta$ between the analytical value calculated for the linear spherocylinder and the estimated value calculated using the convex hull, plotted as a function of the number of particles N .

We added a comment and a figure regarding this point in the Supplementary Information.

Supplementary Information:

Moreover, to further validate our approach, we compare η_{ch} of linear arrangements in the range $N = 1 - 100$ with the theoretical packing fraction of the linear conformation $\eta_{lin} = 2N/(2 + 3(N - 1))$ obtained from the volume of the spherocylinder with N particles, as $\Delta\eta = \eta_{lin} - \eta_{ch}$, as shown in Fig. S7. We find that the precision is again on the fourth decimal digit.

REVIEWERS' COMMENTS

Reviewer #1 (Remarks to the Author):

The authors have addressed in full details all comments in my original review. As a result, it seems to me that the manuscript has gained in clarity and impact.

I am pleased to recommend publication of this manuscript in the present form

Reviewer #2 (Remarks to the Author):

My first review of this manuscript described what I perceived to be the significance of the work, and my opinion of that remains the same. I think that the revised version of the manuscript clarifies what I think were some minor confusions in the original submission. The authors' response addresses my concerns, and I think that this version will be of interest to readers of Nature Communications.